# Hardware Conditioned Policies for Multi-Robot Transfer Learning

**Tao Chen**
The Robotics Institute
Carnegie Mellon University
Pittsburgh, PA 15213
taoc1@cs.cmu.edu

**Adithyavairavan Murali**
The Robotics Institute
Carnegie Mellon University
Pittsburgh, PA 15213
amurali@cs.cmu.edu

**Abhinav Gupta**
The Robotics Institute
Carnegie Mellon University
Pittsburgh, PA 15213
abhinavg@cs.cmu.edu

## Abstract

Deep reinforcement learning could be used to learn dexterous robotic policies but it is challenging to transfer them to new robots with vastly different hardware properties. It is also prohibitively expensive to learn a new policy from scratch for each robot hardware due to the high sample complexity of modern state-of-the-art algorithms. We propose a novel approach called *Hardware Conditioned Policies* where we train a universal policy conditioned on a vector representation of robot hardware. We considered robots in simulation with varied dynamics, kinematic structure, kinematic lengths and degrees-of-freedom. First, we use the kinematic structure directly as the hardware encoding and show great zero-shot transfer to completely novel robots not seen during training. For robots with lower zero-shot success rate, we also demonstrate that fine-tuning the policy network is significantly more sample-efficient than training a model from scratch. In tasks where knowing the agent dynamics is important for success, we learn an embedding for robot hardware and show that policies conditioned on the encoding of hardware tend to generalize and transfer well. Videos of experiments are available at: https://sites.google.com/view/robot-transfer-hcp.

## 1 Introduction

In recent years, we have seen remarkable success in the field of deep reinforcement learning (DRL). From learning policies for games [1, 2] to training robots in simulators [3], neural network based policies have shown remarkable success. But will these successes translate to real world robots? Can we use DRL for learning policies of how to open a bottle, grasping or even simpler tasks like fixturing and peg-insertion? One major shortcoming of current approaches is that they are not sample-efficient. We need millions of training examples to learn policies for even simple actions. Another major bottleneck is that these policies are specific to the hardware on which training is performed. If we apply a policy trained on one robot to a different robot it will fail to generalize. Therefore, in this paradigm, one would need to collect millions of examples for each task and each robot.

But what makes this problem even more frustrating is that since there is no standardization in terms of hardware, different labs collect large-scale data using different hardware. These hardware vary in degrees of freedom (DOF), kinematic design and even dynamics. Because the learning process is so hardware-specific, there is no way to pool and use all the shared data collected across using different types of robots, especially when the robots are trained under torque control. There have been efforts to overcome dependence on hardware properties by learning invariance to robot dynamics using dynamic randomization [4]. However, learning a policy invariant to other hardware properties such as degrees of freedom and kinematic structure is a challenging problem.

In this paper, we propose an alternative solution: instead of trying to learn the invariance to hardware; we embrace these differences and propose to learn a policy conditioned on the hardware properties itself. Our core idea is to formulate the policy $\pi$ as a function of current state $s_t$ and the hardware properties $v_h$. So, in our formulation, the policy decides the action based on current state and its own capabilities (as defined by hardware vector). But how do you represent the robot hardware as vector? In this paper, we propose two different possibilities. First, we propose an explicit representation where the kinematic structure itself is fed as input to the policy function. But such an approach will not be able to encode robot dynamics which might be hard to measure. Therefore, our second solution is to learn an embedding space for robot hardware itself. Our results indicate that encoding the kinematic structures explicitly enables high success rate on zero-shot transfer to new kinematic structure. And learning the embedding vector for hardware implicitly without using kinematics and dynamics information is able to give comparable performance to the model where we use all of the kinematics and dynamics information. Finally, we also demonstrate that the learned policy can also adapt to new robots with much less data samples via finetuning.

## 2 Related Work

**Transfer in Robot Learning** Transfer learning has a lot of practical value in robotics, given that it is computationally expensive to collect data on real robot hardware and that many reinforcement learning algorithms have high sample complexity. Taylor et al. present an extensive survey of different transfer learning work in reinforcement learning [5]. Prior work has broadly focused on the transfer of policies between tasks [6–9], control parameters [10], dynamics [4, 11–13], visual inputs [14], non-stationary environments [15], goal targets [16]. Nilim et. al. presented theoretical results on the performance of transfer under conditions with bounded disturbances in the dynamics [12]. There have been efforts in applying domain adaption such as learning common invariant feature spaces between domains [17] and learning a mapping from target to source domain [18]. Such approaches require prior knowledge and data from the target domain. A lot of recent work has focused on transferring policies trained in simulation to a real robot [4, 14, 19, 20]. However, there has been very limited work on transferring knowledge and skills between different robots [6, 21]. The most relevant paper is Devin et al. [6], who propose module networks for transfer learning and used it to transfer 2D planar policies across hand-designed robots. The key idea is to decompose policy network into robot-specific module and task-specific module. In our work, a universal policy is conditioned on a vector representation of the robot hardware - the policy does not necessarily need to be completely retrained for a new robot. There has also been some concurrent work in applying graph neural networks (GNN) as the policy class in continuous control [22, 23]. [22] uses a GNN instead of a MLP to train a policy. [23] uses a GNN to learn a forward prediction model for future states and performs model predictive control on agents. Our work is orthogonal to these methods as we condition the policy on the augmented state of the robot hardware, which is independant of the policy class.

**Robust Control and Adaptive Control** Robust control can be considered from several vantage points. In the context of trajectory optimization methods, model predictive control (MPC) is a popular framework which continuously resolves an open-loop optimization problem, resulting in a closed loop algorithm that is robust to dynamics disturbances. In the context of deep reinforcement learning, prior work has explored trajectory planning with an ensemble of dynamics models [24], adversarial disturbances [13], training with random dynamics parameters in simulation [4, 11], etc. [4] uses randomization over dynamics so that the policy network can generalize over a large range of dynamics variations for both the robot and the environment. However, it uses position control where robot dynamics has little direct impact on control. We use low-level torque control which is severely affected by robot dynamics and show transfer even between kinematically different agents. There have been similar works in the area of adaptive control [25] as well, where unknown dynamics parameters are estimated online and adaptive controllers adapt the control parameters by tracking motion error. Our work is a model-free method which does not make assumptions like linear system dynamics. We also show transfer results on robots with different DOFs and joint displacements.

**System Identification** System identification is a necessary process in robotics to find unknown physical parameters or to address model inconsistency during training and execution. For control systems based on analytic models, as is common in the legged locomotion community, physical parameters such as the moment of inertia or friction have to be estimated for each custom robotic

hardware [26, 27]. Another form of system identification involves the learning of a dynamics model for use in model-based reinforcement learning. Several prior research work have iterated between building a dynamics model and policy optimization [28–30]. In the context of model-free RL, Yu et al. proposed an Online System Identification [31] module that is trained to predict physical environmental factors such as the agent mass, friction of the floor, etc. which are then fed into the policy along with the agent state [31]. However, results were shown for simple simulated domains and even then it required a lot of samples to learn an accurate regression function of the environmental factors. There is also concurrent work which uses graph networks [23] to learn a forward prediction model for future states and to perform model predictive control. Our method is model-free and only requires a simple hardware augmentation as input regardless of the policy class or DRL algorithms.

## 3 Preliminaries

We consider the multi-robot transfer learning problem under the reinforcement learning framework and deal with fully observable environments that are modeled as continuous space Markov Decision Processes (MDP). The MDPs are represented by the tuple $(\mathcal{S}, \mathcal{A}, P, r, \rho_0, \gamma)$, where $\mathcal{S}$ is a set of continuous states, $\mathcal{A}$ is a set of continuous actions, $P : \mathcal{S} \times \mathcal{A} \times \mathcal{S} \rightarrow \mathbb{R}$ is the transition probability distribution, $r : \mathcal{S} \times \mathcal{A} \rightarrow \mathbb{R}$ is the reward function, $\rho_0$ is the initial state distribution, and $\gamma \in (0, 1]$ is the discount factor. The aim is to find a policy $\pi : \mathcal{S} \rightarrow \mathcal{A}$ that maximizes the expected return.

There are two classes of approaches used for optimization: on-policy and off-policy. On-policy approaches (e.g., Proximal Policy Optimization (PPO) [32]) optimize the same policy that is used to make decisions during exploration. On the other hand, off-policy approaches allow policy optimization on data obtained by a behavior policy different from the policy being optimized. Deep deterministic policy gradient (DDPG) [3] is a model-free actor-critic off-policy algorithm which uses deterministic action policy and is applicable to continuous action spaces.

One common issue with training these approaches is sparse rewards. Hindsight experience replay (HER) [33] was proposed to improve the learning under the sparse reward setting for off-policy algorithms. The key insight of HER is that even though the agent has not succeeded at reaching the specified goal, the agent could have at least achieved a different one. So HER pretends that the agent was asked to achieve the goal it ended up with in that episode at the first place, instead of the one that we set out to achieve originally. By repeating the goal substitution process, the agent eventually learns how to achieve the goals we specified.

## 4 Hardware Conditioned Policies

Our proposed method, *Hardware Conditioned Policies (HCP)*, takes robot hardware information into account in order to generalize the policy network over robots with different kinematics and dynamics. The main idea is to construct a vector representation $v_h$ of each robot hardware that can guide the policy network to make decisions based on the hardware characteristics. Therefore, the learned policy network should learn to act in the environment, conditioned on both the state $s_t$ and $v_h$. There are several factors that encompass robot hardware that we have considered in our framework - robot kinematics (*degree of freedom*, *kinematic structure* such as relative joint positions and orientations, and link length), robot dynamics (*joint damping*, *friction*, *armature*, and *link mass*) - and other aspects such as shape geometry, actuation design, etc. that we will explore in future work. It is also noteworthy that the robot kinematics is typically available for any newly designed robot, for instance through the Universal Robot Description Format-URDF [34]. Nonetheless, the dynamics are typically not available and may be inaccurate or change over time even if provided. We now explain two ways on how to encode the robot hardware via vector $v_h$.

### 4.1 Explicit Encoding

First, we propose to represent robot hardware information via an explicit encoding method (HCP-E). In explicit encoding, we directly use the kinematic structure as input to the policy function. Note that while estimating the kinematic structure is feasible, it is not feasible to measure dynamics. However, some environments and tasks might not be heavily dependent on robot dynamics and in those scenarios explicit encoding (HCP-E) might be simple and more practical than implicit

embedding[1]. We followed the popular URDF convention in ROS to frame our explicit encoding and incorporate the least amount of information to fully define a multi-DOF robot for the explicit encoding. It is difficult to completely define a robot with just its end-effector information, as the kinematic structure (even for the same DOF) affects the robot behaviour. For instance, the whole kinematic chain is important when there are obstacles in the work space and the policy has to learn to avoid collisions with its links.

We consider manipulators composed of $n$ revolute joints $(J_0, J_1, ..., J_{n-1})$. Figure 1 shows two consecutive joints $J_i, J_{i+1}$ on the two ends of an L-shape link and their corresponding local coordinate systems $\{x_i y_i z_i\}$ with origin $O_i$ and $\{x_{i+1} y_{i+1} z_{i+1}\}$ with origin $O_{i+1}$ where $z$-axis is the direction of revolute joint axis. To represent spatial relationship between $J_i, J_{i+1}$, one needs to know the relative pose[2] $P_i$ between $J_i$ and $J_{i+1}$.

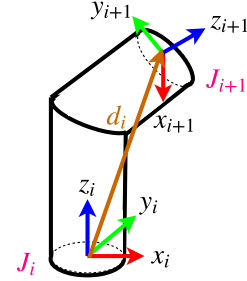

Relative pose $P_i$ can be decomposed into relative position and orientation. Relative position is represented by the difference vector $d_i$ between $O_i$ and $O_{i+1}$, i.e., $d_i = O_{i+1} - O_i$ and $d_i \in \mathbb{R}^3$. The relative rotation matrix from $\{x_{i+1} y_{i+1} z_{i+1}\}$ to $\{x_i y_i z_i\}$ is $\mathbf{R}_i^{i+1} = (\mathbf{R}_w^i)^{-1} \mathbf{R}_w^{i+1}$, where $\mathbf{R}_w^i$ is the rotation matrix of $\{x_i y_i z_i\}$ relative to the world coordinate system. One can further convert rotation matrix which has 9 elements into Euler rotation vector with only 3 independent elements. Therefore, relative rotation can be represented by an Euler rotation vector $e_i = (\theta_{ix}, \theta_{iy}, \theta_{iz}), e_i \in \mathbb{R}^3$. The relative pose is then $P_i = d_i \oplus e_i \in \mathbb{R}^6$, where $\oplus$ denotes concatenation.

Figure 1: Local coordinate systems for two consecutive joints

With relative pose $P_i$ of consecutive joints in hand, the encoding vector $v_h$ to represent the robot can be explicitly constructed as follows[3]:

$$v_h = P_{-1} \oplus P_0 \oplus \cdots \oplus P_{n-1}$$

## 4.2  Implicit Encoding

In the above section, we discussed how kinematic structure of the hardware can be explicitly encoded as $v_h$. However, in most cases, we need to not only encode kinematic structure but also the underlying dynamic factors. In such scenarios, explicit encoding is not possible since one cannot measure friction or damping in motors so easily. In this section, we discuss how we can learn an embedding space for robot hardware while simultaneously learning the action policies. Our goal is to estimate $v_h$ for each robot hardware such that when a policy function $\pi(s_t, v_h)$ is used to take actions it maximizes the expected return. For each robot hardware, we initialize $v_h$ randomly. We also randomly initialize the parameters of the policy network. We then use standard policy optimization algorithms to update network parameters via back-propagation. However, since $v_h$ is also a learned parameter, the gradients flow back all the way to the encoding vector $v_h$ and update $v_h$ via gradient descent: $v_h \leftarrow v_h - \alpha \nabla_{v_h} L(v_h, \theta)$, where $L$ is the cost function, $\alpha$ is the learning rate. Intuitively, HPC-I trains the policy $\pi(s_t, v_h)$ such that it not only learns a mapping from states to actions that maximizes the expected return, but also finds a good representation for the robot hardware simultaneously.

## 4.3  Algorithm

The hardware representation vector $v_h$ can be incorporated into many deep reinforcement learning algorithms by augmenting states to be: $\hat{s}_t \leftarrow s_t \oplus v_h$. We use PPO in environments with dense reward and DDPG + HER in environments with sparse reward in this paper. During training, a robot will be randomly sampled in each episode from a pre-generated robot pool $\mathcal{P}$ filled with a large number of robots with different kinematics and dynamics. Alg. 1 provides an overview of our algorithm. The detailed algorithms are summarized in Appendix A (Alg. 2 for on-policy and Alg. 3 for off-policy).

**Algorithm 1** Hardware Conditioned Policies (HCP)

---

Initialize a RL algorithm $\Psi$                $\triangleright$ e.g. PPO, DDPG, DDPG+HER
Initialize a robot pool $\mathcal{P}$ of size $N$ with robots in different kinematics and dynamics
**for** episode = 1:M **do**
    Sample a robot instance $\mathcal{I} \in \mathcal{P}$
    Sample an initial state $s_0$
    Retrieve the robot hardware representation vector $v_h$
    Run policy $\pi$ in the environment for $T$ timesteps
    Augment all states with $v_h$: $\hat{s} \leftarrow s \oplus v_h$          $\triangleright \oplus$ denotes concatenation
    **for** n=1:W **do**
        Optimize actor and critic networks with $\Psi$ via minibatch gradient descent
        **if** $v_h$ is to be learned (i.e. for implicit encoding, HCP-I) **then**
            update $v_h$ via gradient descent in the optimization step as well
        **end if**
    **end for**
**end for**

---

## 5 Experimental Evaluation

Our aim is to demonstrate the importance of conditioning the policy based on a hardware representation $v_h$ for transferring complicated policies between dissimilar robotic agents. We show performance gains on two diverse settings of manipulation and hopper.

### 5.1 Explicit Encoding

**Robot Hardwares:** We created a set of robot manipulators based on the Sawyer robot in MuJoCo [35]. The basic robot types are shown in Figure 2. By permuting the chronology of revolute joints and ensuring the robot design is feasible, we designed 9 types of robots (named as A, B,..., I) in which the first four are 5 DOF, the next four are 6 DOF, and the last one is 7 DOF, following the main feasible kinematic designs described in hardware design literature [36]. Each of these 9 robots were further varied with different link lengths and dynamics.

**Tasks:** We consider reacher and peg insertion tasks to demonstrate the effectiveness of explicit encoded representation. In reacher, robot starts from a random initial pose and it needs to move the end effector to the random target position. In peg-insertion, a peg is attached to the robot gripper and the task is to insert the peg into the hole on the table. It's considered a success only if the peg bottom goes inside the hole more than $0.03$m. Goals are described by the 3D target positions $(x_g, y_g, z_g)$ of end effector (reacher) or peg bottom (peg insertion).

**States and Actions**: The states of both environments consist of the angles and velocities of all robot joints. Action is $n$-dimensional torque control over $n$ ($n \leq 7$) joints. Since we consider robots with different DOF in this paper, we use zero-padding for robots with $< 7$ joints to construct a fixed-length state vector for different robots. And the policy network always outputs 7-dimensional actions, but only the first $n$ elements are used as the control command.

**Robot Representation**: As mentioned in section 4.1, $v_h$ is explicitly constructed to represent the robot kinematic chain: $v_h =$

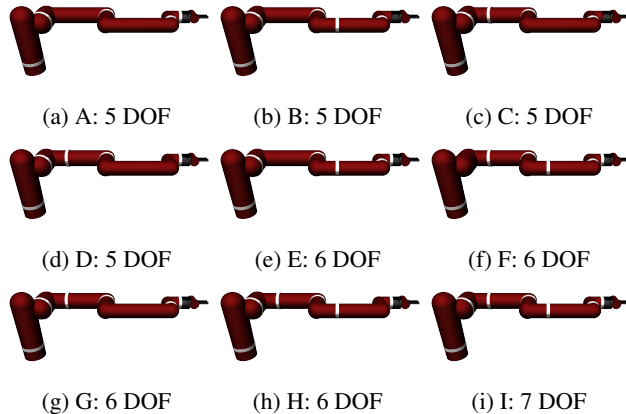

(a) A: 5 DOF     (b) B: 5 DOF     (c) C: 5 DOF

(d) D: 5 DOF     (e) E: 6 DOF     (f) F: 6 DOF

(g) G: 6 DOF     (h) H: 6 DOF     (i) I: 7 DOF

Figure 2: Robots with different DOF and kinematics structures. The white rings represent joints. There are 4 variants of 5 and 6 DOF robots due to the different placements of joints.

$P_{-1} \oplus P_0 \oplus \cdots \oplus P_{n-1}$. We use zero-padding for robots with $< 7$ joints to construct a fixed-length representation vector $v_h$ for different robots.

**Rewards**: We use binary sparse reward setting because sparse reward is more realistic in robotics applications. And we use DPPG+HER as the backbone training algorithm. The agent only gets $+1$ reward if POI is within $\epsilon$ euclidean distance of the desired goal position. Otherwise, it gets $-1$ reward. We use $\epsilon = 0.02$m in all experiments. However, this kind of sparse reward setting encourages the agent to complete the task using as less time steps as possible to maximize the return in an episode, which encourages the agent to apply maximum torques on all the joints so that the agent can move fast. This is referred to as bang–bang control in control theory[37]. Hence, we added action penalty on the reward.

More experiment details are shown in Appendix B.

### 5.1.1  Does HCP-E improve performance?

To show the importance of hardware information as input to the policy network, we experiment on learning robotic skills among robots with different dynamics (joint damping, friction, armature, link mass) and kinematics (link length, kinematic structure, DOF). The 9 basic robot types are listed in Figure 2. We performed several leave-one-out experiments (train on 8 robot types, leave 1 robot type untouched) on these robot types. The sampling ranges for link length and dynamics parameters are shown in Table 2 in Appendix B.1.1. We compare our algorithm with vanilla DDPG+HER (trained with data pooled from all robots) to show the necessity of training a universal policy network conditioned on hardware characteristics. Figure 3 shows the learning curves[4] of training on robot types A-G and I. It clearly shows that our algorithm HCP-E outperforms the baseline. In fact, DDPG+HER without any hardware information is unable to learn a common policy across multiple robots as different robots will behave differently even if they execute the same action in the same state. More leave-one-out experiments are shown in Appendix C.4.

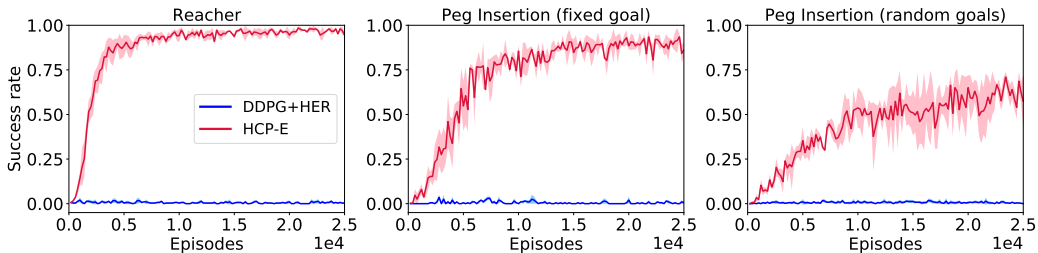

Figure 3: Learning curves for multi-DOF setup. Training robots contain Type A-G and Type I robots (four 5-DOF types, three 6-DOF types, one 7-DOF type). Each type has 140 variants with different dynamics and link lengths. The 100 testing robots used to generate the learning curves are from the same training robot types but with different link lengths and dynamics. (a): reacher task with random initial pose and target position. (b): peg insertion with fixed hole position. (c): peg insertion with hole position $(x, y, z)$ randomly sampled in a $0.2$m box region. Notice that the converged success rate in (c) is only about $70\%$. This is because when we randomly generate the hole position, some robots cannot actually insert the peg into hole due to physical limit. Some hole positions are not inside the reachable space (workspace) of the robots. This is especially common in 5-DOF robots.

### 5.1.2  Is HCP-E capable of zero-shot transfer to unseen robot kinematic structure?

We now perform testing in the leave-one-out experiments. Specifically, we can test the zero-shot transfer ability of policy network on new type of robots. Table 1 shows the quantitative statistics about testing performance on new robot types that are different from training robot types. Each data[5] in the table is obtained by running the model on 1000 unseen test robots (averaged over 10 trials, 100 robots per trial) of that robot type but with different link lengths and dynamics.

Table 1: Zero-shot testing performance on new robot type

| Exp. | Tasks | Training Robot Types | Testing Robot Type | Alg. | Success rate (%) |
|------|-------|----------------------|--------------------|------|------------------|
| I | Reacher (random goals) | A-G + I | H | HCP-E | **92.50 ± 1.96** |
| II | | | | DDPG+HER | 0.20 ± 0.40 |
| III | | A-D + F-I | E | HCP-E | **88.00 ± 2.00** |
| IV | | | | DDPG+HER | 2.70 ± 2.19 |
| V | Peg Insertion (fixed goal) | A-G + I | H | HCP-E | **92.20 ± 2.75** |
| VI | | | | DDPG+HER | 0.00 ± 0.00 |
| VII | | A-D + F-I | E | HCP-E | **87.60 ± 2.01** |
| VIII | | | | DDPG+HER | 0.80 ± 0.60 |
| IX | | A-H | I | HCP-E | **65.60 ± 3.77** |
| X | | | | DDPG+HER | 0.10 ± 0.30 |
| XI | Peg Insertion (random goals) | A-G + I | H | HCP-E | **4.10 ± 1.50** |
| XII | | | | DDPG+HER | 0.10 ± 0.30 |
| XIII | | A-D + F-I | E | HCP-E | **76.10 ± 3.96** |
| XIV | | | | DDPG+HER | 0.00 ± 0.00 |
| XV | | A-H | I | HCP-E | **23.50 ± 4.22** |
| XVI | | | | DDPG+HER | 0.20 ± 0.40 |

From Table 1, it is clear that HCP-E still maintains high success rates when the policy is applied to new types of robots that have never been used in training, while DDPG+HER barely succeeds at controlling new types of robots at all. The difference between using robot types A-G+I and A-D+F-I (both have four 5-DOF types, three 6-DOF types, and one 7-DOF type) is that robot type H is harder for peg insertion task than robot type E due to its joint configuration (it removes joint $J_5$). As we can see from Exp. I and V, HCP-E got about $90\%$ zero-shot transfer success rate even if it's applied on the hard robot type H. Exp. X and XVI show the model trained with only 5 DOF and 6 DOF being applied to 7-DOF robot type. We can see that it is able to get about $65\%$ success rate in peg insertion task with fixed goal[6].

**Zero-shot transfer to a real Sawyer robot:** We show results on the multi-goal reacher task, as peg insertion required additional lab setup. Though the control frequency on the real robot is not as stable as that in simulation, we still found a high zero-shot transfer rate. For quantitative evaluation, we ran three policies on the real robot with results averaged over 20 random goal positions. **A** used the policy from Exp. I (HCP-E), **B** used the policy from Exp. II (DDPG+HER) while **C** used the policy trained with actual Sawyer CAD model in simulation with just randomized dynamics. The distance from target for the 20-trials are summarized in Figure 4. Despite of the large reality gap[7], HCP-E (BLUE) is able to reach the target positions with a high success rate ($75\%$) [8]. DDPG+HER (RED) without hardware information was not even able to move the arm close to the desired position.

**Fine-tuning the zero-shot policy:** Table 1 also shows that Exp. XI and Exp. XV have relatively low zero-shot success rates on new type of robots. Exp. XI is trained on easier 6-DOF robots

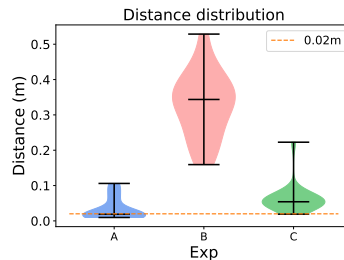

Figure 4: Testing distance distribution on a real sawyer robot. **A** used the policy from Exp. I, **B** used the policy from Exp. II, **C** used the policy trained with the actual Sawyer CAD model in simulation with randomized dynamics.

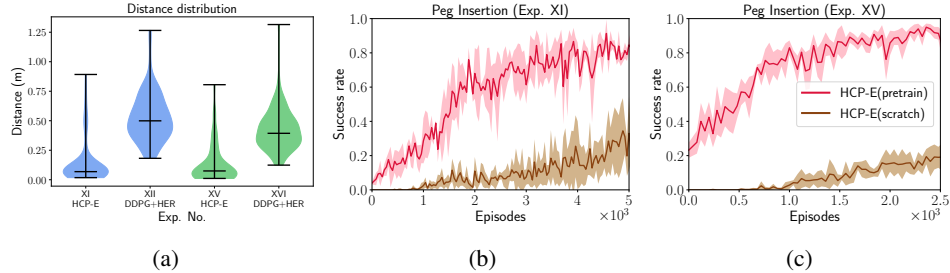

(a)    (b)    (c)

Figure 5: (a): Distribution (violin plots) of distance between the peg bottom at the end of episode and the desired position. The three horizontal lines in each violin plot stand for the lower extrema, median value, and the higher extrema. It clearly shows that HCP-E moves the pegs much closer to the hole than DDPG+HER. (b): The brown curve is the learning curve of training HCP-E on robot type H with different link lengths and dynamics in multi-goal setup from scratch. The pink curve is the learning curve of training HCP-E on same robots with pretrained model from Exp. XI. (c): Similar to (b), the training robots are robot type I (7 DOF) and the pretrained model is from Exp. XV. (b) and (c) show that applying the pretrained model that is trained on different robot types to a new robot type can accelerate the learning by a large margin.

(E, F, G) and applied to a harder 6-DOF robot type (H). Exp. XV is trained only on 5-DOF and 6-DOF robots and applied to 7-DOF robots (I). The hole positions are randomly sampled in both experiments. Even though the success rates are low, HCP-E is actually able to move the peg bottom close to the hole in most testing robots, while DDPG+HER is much worse, as shown in Figure 5a. We also fine-tune the model specifically on the new robot type for these two experiments, as shown in Figure 5b and Figure 5c. It's clear that even though zero-shot success rates are low, the model can quickly adapt to the new robot type with the pretrained model weights.

## 5.2  Implicit Encoding

**Environment** HCP-E shows remarkable success on transferring manipulator tasks to different types of robots. However, in explicit encoding we only condition on the kinematics of the robot hardware. For unstable systems in robotics, such as in legged locomotion where there is a lot of frequent nonlinear contacts [13], it is crucial to consider robot dynamics as well. We propose to learn an implicit, latent encoding (HCP-I) for each robot without actually using any kinematics and dynamics information. We evaluate the effectiveness of HCP-I on the 2D hopper [38]. Hopper is an ideal environment as it is an unstable system with sophisticated second-order dynamics involving contact with the ground. We will demonstrate that adding implicitly learned robot representation can lead to comparable performance to the case where we know the ground-truth kinematics and dynamics. To create robots with different kinematics and dynamics, we varied the length and mass of each hopper link, damping, friction, armature of each joint, which are shown in Table 5 in Appendix B.

**Performance** We compare HCP-I with HCP-E, HCP-E+ground-truth dynamics, and vanilla PPO model without kinematics and dynamics information augmented to states. As shown in Figure 6a, HCP-I outperforms the baseline (PPO) by a large margin. In fact, with the robot representation $v_h$ being automatically learned, we see that HCP-I achieves comparable performance to HCP-E+Dyn that uses both kinematics and dynamics information, which means the robot representation $v_h$ learns the kinematics and dynamics implicitly. Since dynamics plays a key role in the hopper performance which can be seen from the performance difference between HCP-E and HCP-E+Dyn, the implicit encoding method obtains much higher return than the explicit encoding method. This is because the implicit encoding method can automatically learn a good robot hardware representation and include the kinematics and dynamics information, while the explicit encoding method can only encode the kinematics information as dynamics information is generally unavailable.

**Transfer Learning on New Agents** We now apply the learned HCP-I model as a pretrained model onto new robots. However, since $v_h$ for the new robot is unknown, we fine-tune the policy parameters and also estimate $v_h$. As shown in Figure 6b, HCP-I with pretrained weights learns much faster than HCP-I trained from scratch. While in the current version, we do not show explicit few-shot results,

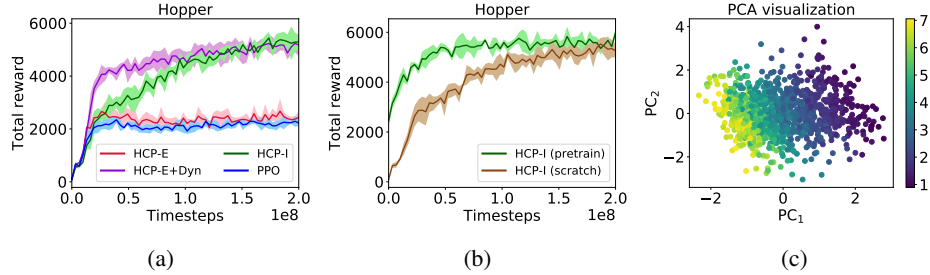

|     |     |     |
| --- | --- | --- |
| (a) | (b) | (c) |

Figure 6: (a): Learning curves using 1000 hoppers with different kinematics and dynamics in training. HCP-I is able to automatically learn a good robot representation such that the learning performance can be on par with HCP-E+Dyn where we use the ground-truth kinematics and dynamics values. And HCP-I has a much better performance than vanilla PPO. (b): If we use the pretrained HCP-I model (only reuse the hidden layers) from (a) on 100 new hoppers, HCP-I with pretrained weights learns much faster than training from scratch. (c): Embedding visualization. The colorbar shows the hopper torso mass value. We can see that the embedding is smooth as the color transition is smooth.

one can train a regression network to predict the trained $v_h$ based on the agent's limited interaction with environment.

**Embedding Smoothness** We found the implicit encoding $v_h$ to be a smooth embedding space over the dynamics. For example, we only vary one parameter (the torso mass) and plot the resultant embedding vectors. Notice that we reduce the dimension of $v_h$ to 2 since we only vary torso mass. Figure 6c shows a smooth transition over torso mass (the color bar represents torso mass value, 1000 hoppers with different torso mass), where robots with similar mass are clustered together.

## 6    Conclusion

We introduced a novel framework of *Hardware Conditioned Policies* for multi-robot transfer learning. To represent the hardware properties as a vector, we propose two methods depending on the task: explicit encoding (HCP-E) and implicit encoding (HCP-I). HCP-E works well when task policy does not heavily depend on agent dynamics. It has an obvious advantage that it is possible to transfer the policy network to new robots in a zero-shot fashion. Even when zero-shot transfer gives low success rate, we showed that HCP-E actually brings the agents very close to goals and is able to adapt to the new robots very quickly with finetuning. When the robot dynamics is so complicated that feeding dynamics information into policy network helps improve learning, the explicit encoding is not enough as it can only encode the kinematics information and dynamics information is usually challenging and sophisticated to acquire. To deal with such cases, we propose an implicit encoding scheme (HCP-I) to learn the hardware embedding representation automatically via back-propagation. We showed that HCP-I, without using any kinematics and dynamics information, can achieve good performance on par with the model that utilized both ground truth kinematics and dynamics information.

## Acknowledgement

This research is partly sponsored by ONR MURI N000141612007 and the Army Research Office and was accomplished under Grant Number W911NF-18-1-0019. Abhinav was supported in part by Sloan Research Fellowship and Adithya was partly supported by a Uber Fellowship. The views and conclusions contained in this document are those of the authors and should not be interpreted as representing the official policies, either expressed or implied, of the Army Research Office or the U.S. Government. The U.S. Government is authorized to reproduce and distribute reprints for Government purposes notwithstanding any copyright notation herein. The authors would also like to thank Senthil Purushwalkam and Deepak Pathak for feedback on the early draft and Lerrel Pinto and Roberto Shu for insightful discussions.

## Footnotes

[1]We will elaborate such environments on section 5.1. We also experimentally show the effect of dynamics for transferring policies in such environments in Appendix C.1 and C.2.

[2]If the manipulators only differ in length of all links, $v_h$ can be simply the vector of each link's length. When the kinematic structure and DOF also vary, $v_h$ composed of link length is not enough.

[3]$P_i$ is relative pose from $U$ to $V$. If $i = -1$, $U = J_0$, $V =$ robot base. If $i = 0, 1, ..., n-2$, $U = J_{i+1}$, $V = J_i$. If $i = n - 1$, $U =$ end effector, $V = J_{n-1}$.

[4]The learning curves are averaged over 5 random seeds on 100 testing robots and shaded areas represent 1 standard deviation.

[5]The success rate is represented by the mean and standard deviation

[6]Since we are using direct torque control without gravity compensation, the trivial solution of transferring where the network can regard the 7-DOF robot as a 6-DOF robot by keeping one joint fixed doesn't exist here.

[7]The reality gap is further exaggerated by the fact that we didn't do any form of gravity compensation in the simulation but the real-robot tests used the gravity compensation to make tests safer.

[8]The HCP-E policy resulted in a motion that was jerky on the real Sawyer robot to reach the target positions. This was because we used sparse reward during training. This could be mitigated with better reward design to enforce smoothness.

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
