[Supplementary Material]

# Appendix A    Algorithms

## A.1    Algorithms

In this section, we present two detailed practical algorithms based on the HCP concept. Alg. 2 is HCP based on PPO which can be used to solve tasks with dense reward. Alg. 3 is HCP based on DDPG+HER which can be used to solve multi-goal tasks with sparse reward.

---

**Algorithm 2** Hardware Conditioned Policy (HCP) - on-policy

---

Initialize PPO algorithm
Initialize a robot pool $\mathcal{P}$ of size $N$ with robots in different dynamics and kinematics
**for** episode = 1, M  **do**
    **for** actor=1, K **do**
        Sample a robot instance $\mathcal{I} \in \mathcal{P}$
        Sample an initial state $s_0$
        Retrieve the robot hardware representation vector $v_h$
        Augment $s_0$:
$$\hat{s}_0 \leftarrow s_0 \oplus v_h$$
        **for** t = 0, T-1 **do**
            Sample action $a_t \leftarrow \pi(\hat{s}_t)$ using current policy
            Execute action $a_t$, receive reward $r_t$, observe new state $s_{t+1}$, and augmented state $\hat{s}_{t+1}$
        **end for**
        Compute advantage estimates $A_0, A_1, ..., A_{T-1}$
    **end for**
    **for** n=1,W **do**
        Optimize actor and critic networks with PPO via minibatch gradient descent
        **if** $v_h$ is to be learned **then**
            update $v_h$ via gradient descent in the optimization step as well
        **end if**
    **end for**
**end for**

---

**Algorithm 3** Hardware Conditioned Policy (HCP) - off-policy

---

Initialize DDPG algorithm
Initialize experience replay buffer $\mathcal{R}$
Initialize a robot pool $\mathcal{P}$ of size $N$ with robots in different dynamics and kinematics
**for** episode = 1, M  **do**
    Sample a robot instance $\mathcal{I} \in \mathcal{P}$
    Sample a goal position $g$ and an initial state $s_0$
    Retrieve the robot hardware representation vector $v_h$
    Augment $s_0$:
$$\hat{s}_0 \leftarrow s_0 \oplus g \oplus v_h$$
    **for** t = 0, T-1 **do**
        Sample action $a_t \leftarrow \pi_b(\hat{s}_t)$ using behavioral policy
        Execute action $a_t$, receive reward $r_t$, observe new state $s_{t+1}$, and augmented state $\hat{s}_{t+1}$
        Store $(\hat{s}_t, a_t, r_t, \hat{s}_{t+1})$ into $\mathcal{R}$
    **end for**
    Augment $\mathcal{R}$ with pseudo-goals via HER
    **for** n=1,W **do**
        Optimize actor and critic networks with DDPG via minibatch gradient descent
        **if** $v_h$ is to be learned **then**
            update $v_h$ via gradient descent in the optimization step as well
        **end if**
    **end for**
**end for**

---

# Appendix B Experiment Details

We performed experiments on three environments in this paper: reacher, peg insertion, and hopper, as shown in Figure 7. Videos of experiments are available at: `https://sites.google.com/view/robot-transfer-hcp`.

## B.1 Reacher and Peg Insertion

The reason why we choose reacher and peg insertion task is that most of manipulator tasks like welding, assembling, grasping can be seen as a sequence of reacher tasks in essence. Reacher task is the building block of many manipulator tasks. And peg insertion task can further show the control accuracy and robustness of the policy network in transferring torque control to new robots.

(a) reacher    (b) peg insertion    (c) hopper

Figure 7: (a): reacher, the green box represents end effector initial position distribution, and the yellow box represents end effector target position distribution. (b): peg insertion. The white rings in (a) and (b) represent joints. (c): hopper.

### B.1.1 Robot Variants

During training time, we consider 9 basic robot types (named as Type A,B,...,I) as shown in Figure 2 which have different DOF and joint placements. The 5-DOF and 6-DOF robots are created by removing joints from the 7-DOF robot.

We also show the length range of each link and dynamics parameter ranges in Table 2. The link name and joint name conventions are defined in Figure 8. Notice that damping values ranged from $[0, 1), (1, +\infty)$ are called underdamped and overdamped systems respectively. As these systems have very different dynamics characteristics, $50\%$ of the damping values sampled are less than 1, and the rest $50\%$ are greater than or equal to 1.

Figure 8: Link name and joint name convention

Table 2: Manipulator Parameters

| Kinematics | |
|---|---|
| Links | Length Range (m) |
| $l_0$ | $0.290 \pm 0.10$ |
| $l_{1\_1}$ | $0.119 \pm 0.05$ |
| $l_{1\_2}$ | $0.140 \pm 0.07$ |
| $l_2$ | $0.263 \pm 0.12$ |
| $l_{3\_1}$ | $0.120 \pm 0.06$ |
| $l_{3\_2}$ | $0.127 \pm 0.06$ |
| $l_4$ | $0.275 \pm 0.12$ |
| $l_{5\_1}$ | $0.096 \pm 0.04$ |
| $l_{5\_2}$ | $0.076 \pm 0.03$ |
| $l_6$ | $0.049 \pm 0.02$ |
| Dynamics | |
| damping | $[0.01, 30]$ |
| friction | $[0, 10]$ |
| armature | $[0.01, 4]$ |
| link mass | $[0.25, 4] \times$ default mass |

Even though we only train with robot types listed in Figure 2, our policy can be directly transferred to other new robots like the Fetch robot shown in Figure 9.

### B.1.2 Hyperparameters

We closely followed the settings in original DDPG paper. Actions were added at the second hidden layer of $Q$. All hidden layers used scaled exponential linear unit (SELU) as the activation function and we used Adam optimizer. Other hyperparameters are summarized in Table 3.

(a) J: new 5-DOF robot

(b) K: new 7-DOF robot (Fetch)

Figure 9: New types of robot. We used the model trained in Exp. V and directly applied to robot shown in (a) and Fetch robot shown in (b). We tested the model on 1000 unseen test robots (averaged over 10 trials, 100 robots per trial) of type J (as shown in (a)), and got $86.30 \pm 4.41\%$ success rate. We tested on the fetch robot in (b) 10 times and got $100\%$ success rate.

**Initial position distributions**: For reacher task, the initial position of end effector is randomly sampled from a box region $0.3\text{m} \times 0.4\text{m} \times 0.2\text{m}$. For peg insertion task, all robots start from a horizontal fully-expanded pose.

**Goal distributions**: For reacher task, the target end effector position region is a box region $0.3\text{m} \times 0.6\text{m} \times 0.4\text{m}$ which is located $0.2\text{m}$ below the initial position sampling region. For peg insertion task, we have experiments on hole position fixed and hole position randomly moved. If the hole position is to be randomly moved, the table's position will be randomly sampled from a box region $0.2\text{m} \times 0.2\text{m} \times 0.2\text{m}$.

**Rewards**: As mentioned in paper, we add action penalty on rewards so as to avoid bang-bang control. The reward is defined as: $r(s_t, a_t, g) = \text{sgn}_{\pm}(\epsilon - \|s_{t+1}(\text{POI}) - g\|_2) - \beta \|a_t\|_2^2$, where $s_{t+1}(\text{POI})$ is the position of the point of interest (POI, end effector in reacher and peg bottom in peg insertion) after the execution of action $a_t$ in the state $s_t$, $\beta$ is a hyperparameter $\beta > 0$ and $\beta \|a_t\|_2^2 \ll 1$.

**Success criterion**: For reacher task, the end effector has to be within $0.02\text{m}$ Euclidean distance to the target position to be considered as a success. For peg insertion task, the peg bottom has to be within $0.02\text{m}$ Euclidean distance to the target peg bottom position to be considered as a success. Since the target peg bottom position is always $0.05\text{m}$ below the table surface no matter how table moves, so the peg has to be inserted into the hole more than $0.03\text{m}$.

**Observation noise**: We add uniformly distributed observation noise on states (joint angles and joint velocities). The noise is uniformly sampled from $[-0.02, 0.02]$ for both joint angles ($\text{rad}$) and joint velocities ($\text{rad}\,\text{s}^{-1}$).

Table 3: Hyperparameters for reacher and peg insertion tasks

| | |
|---|---|
| number of training robots for each type | 140 |
| success distance threshold $\epsilon$ | 0.02m |
| maximum episode time steps | 200 |
| actor learning rate | 0.0001 |
| critic learning rate | 0.0001 |
| critic network weight decay | 0.001 |
| hidden layers | 128-256-256 |
| discount factor $\gamma$ | 0.99 |
| batch size | 128 |
| warmup episodes | 50 |
| experience replay buffer size | 1000000 |
| network training iterations after each episode | 100 |
| soft target update $\tau$ | 0.01 |
| number of future goals $k$ | 4 |
| action penalty coefficient $\beta$ | 0.1 |
| robot control frequency | 50Hz |

Table 4: Hyperparameters for hopper

| | |
|---|---|
| number of training hoppers | 1000 |
| number of actors K | 8 |
| maximum episode time steps | 2048 |
| learning rate | 0.0001 |
| hidden layers | 128-128 |
| discount factor $\gamma$ | 0.99 |
| GAE parameter $\lambda$ | 0.95 |
| clip ratio $\eta$ | 0.2 |
| batch size | 512 |
| $v_h$ dimension | 32 |
| network training epochs after each rollout | 5 |
| value function loss coefficient $c_1$ | 0.5 |
| entropy loss coefficient $c_2$ | 0.015 |

## B.2 Hopper

We used the same reward design as the hopper environment in OpenAI Gym. As it's a dense reward setting, we use PPO for this task. All hidden layers used scaled exponential linear unit (SELU) as the activation function and we used Adam optimizer. Other hyperparameters are summarized in Table 4. And the sampling ranges of link lengths and dynamics are shown in Table 5.

# Appendix C Supplementary Experiments

In section C.1 and section C.2, we explore the dynamics effect in manipulators. Section C.3 shows

Table 5: Hopper Parameters

| Kinematics | |
|---|---|
| Links | Length Range (m) |
| torso | $0.40 \pm 0.10$ |
| thigh | $0.45 \pm 0.10$ |
| leg | $0.50 \pm 0.15$ |
| foot | $0.39 \pm 0.10$ |
| Dynamics | |
| damping | $[0.01, 5]$ |
| friction | $[0, 2]$ |
| armature | $[0.1, 2]$ |
| link mass | $[0.25, 2] \times$ default mass |

the learning curves for 7-DOF robots with different link lengths and dynamics. In section C.4, we show more training details of HCP-E experiments on different combinations of robot types and how well HCP-E models perform on robots that belong to the same training robot types but with different link lengths and dynamics.

## C.1 Effect of Dynamics in Transferring Policies for Manipulation

Explicit encoding is made possible when knowing the dynamics of the system doesn't help learning. In such environments, as long as the policy network is exposed to a diversity of robots with different dynamics during training, it will be robust enough to adapt to new robots with different dynamics. To show that knowing ground-truth dynamics doesn't help training for reacher and peg insertion tasks, we experimented on 7-DOF robots (Type I) with different dynamics only with following algorithms:DDPG+HER, DDPG+HER+dynamics, DDPG+HER+random number. The first one uses DDPG+HER with only joint angles and joint velocities as the state. The second experiment uses DDPG+HER with the dynamics parameter vector added to the state. The dynamics are scaled to be within $[0, 1)$. The third experiment uses DDPG+HER with a random vector ranged from $[0, 1)$ added to states that is of same size as the dynamics vector. The dynamics parameters sampling ranges are shown in Table 2. The number of training robots is 100.

Figure 10 shows that DDPG+HER with only joint angles and joint velocities as states is able to achieve about $100\%$ success rate in both reacher and peg insertion (fixed hole position) tasks. In fact, we see that even if state is augmented with a random vector, the policy network can still generalize over new testing robots, which means the policy network learns to ignore the augmented part. Figure 10 also shows that with ground-truth dynamics parameters or random vectors input to the policy and value networks, the learning process becomes slower. In hindsight, this makes sense because the dynamics information is not needed for the policy network and if we forcefully feed in those information, it will take more time for the network to learn to ignore this part and train a robust policy across robots with different dynamics.

Figure 10: Learning curves on 7-DOF robots with different dynamics only.

## C.2 How robust is the policy network to changes in dynamics?

We performed a stress test on the generalization or robustness of the policy network to variation in dynamics. The experiments are similar to those in section C.1, but the training joint damping values are randomly sampled from $[0.01, 2)$ this time. Other dynamics parameters are still randomly sampled according to Table 2. The task here is peg insertion. Figure 11 and Table 6 show the generalization capability of the DDPG + HER model with only joint angles and joint velocities as the state.

We can see from Figure 11 and Table 6 that even though the DDPG + HER model is trained with joint damping values sampled from $[0.01, 2)$, it can successfully control robots with damping values sampled from other ranges including $[2, 10), [10, 20), [20, 30)$ with $100\%$ success rate. It is noteworthy that a damping value of 1 corresponds to critical damping (which is what most practical systems aim for), while $< 1$ is under-damped and above is over-damped. For the damping range $[30, 40)$, the success rate is

Table 6: Success rate on 100 testing robots

| Testing damping range | Success rate |
|---|---|
| $[0.01, 2)$ | $100\%$ |
| $[2, 10)$ | $100\%$ |
| $[10, 20)$ | $100\%$ |
| $[20, 30)$ | $100\%$ |
| $[30, 40)$ | $85\%$ |
| $[40, 50)$ | $47\%$ |

$85\%$. In damping range $[40, 50)$, the success rate is $47\%$. Note that each joint has a torque limit, so when damping becomes too large, the control is likely to be unable to move some joints and thus fail. Also, the larger the damping values are, the more time steps it takes to finish the peg insertion task, as shown in Figure 11b.

(a) Distance to desired position at last step          (b) Episode length

Figure 11: Performance (violin plots) on 100 testing robots with damping values sampled from different ranges using the DDPG+HER model trained with damping range $[0.01, 2)$). Other dynamics parameters are still randomly sampled according to Table 2. The left plot shows the distribution of the distances between the robot's peg bottom and the target peg bottom position at the end of episode. The right plot shows the distribution of the episode length. An episode will be ended early if the peg is inserted into the hole successfully and the maximum number of episode time steps is 200. The three horizontal lines in each violin plot stand for the lower extrema, median value, and the higher extrema.

## C.3 Learning curves for 7-DOF robots with different link length and dynamics

In this section, we provide two supplementary experiments on training 7-DOF (type I) to perform reacher and peg insertion task.

Figure 12: Learning curves for 7-DOF robots with different link length and dynamics. We show the HCP-E+Dyn learning curves only for comparison. In real robots, dynamics parameters are usually not easily accessible. So it's not pratical to use dynamics information in robotics applications. We can see that both HCP-I and HCP-E got much higher success rates on both tasks than vanilla DDPG+HER.

## C.4   Multi-DOF robots learning curves

Figure 13 provides more details of training progress in different experiments.

Figure 13: Learning curves for multi-DOF setting. Symbol A,B,...,I in the figure represent the types of robot used in training. All these experiments are only trained on 8 types of robots (leave one out). The 100 testing robots used to generate the learning curves are from the same training robot types but with different link length and dynamics. The second row shows the results on peg insertion task with hole position randomly generated within a $0.2m$ box region. (a): reacher task with robot types A-D + F-I. (b): peg insertion task with a fixed hole position with robot types A-D + F-I. (c): peg insertion task with a fixed hole position with robot types A-H. (d): peg insertion task with a random hole position with robot types A-G + I. (e): peg insertion task with a random hole position with robot types A-D + F-I. (f): peg insertion task with a random hole position with robot types A-H.

Table 1 in the paper shows how well HCP-E models perform when they are applied to the new robot type that has never been trained before. Table 7 to 14 show how the universal policy behaves on the robot types that have been trained before. These robots are from the training robot types, but with different link lengths and dynamics.

The less DOF the robot has, the less dexterous the robot can be. Also, where to place the $n$ joints affects the workspace of the robot and determine how flexible the robot can be. Therefore, we can see some low success rate data even in trained robot types. For example,the trained HCP-E model only got 6.70 success rate when tested on robot type D which has actually been trained in peg insertion tasks with random hole positions, as shown in Table 12. This is because its joint displacements and number of DOFs limit the flexibility as shown in Figure 2d. Type D doesn't have joint $J_4$ and $J_5$ which are crucial for peg insertion tasks.

Table 7: Zero-shot testing performance on training robot types (Exp. I & II)

| Alg. | Testing Robot Types | | | | | | | |
|---|---|---|---|---|---|---|---|---|
| | A | B | C | D | E | F | G | I |
| HCP-E | 93.10± 2.91 | 95.70± 1.55 | 98.20± 1.55 | 97.50± 1.02 | 95.30± 1.49 | 94.00± 3.26 | 98.40± 1.11 | 97.90± 1.67 |
| DDPG+HER | 1.00± 1.22 | 1.00± 1.00 | 2.50± 1.36 | 0.10± 0.30 | 0.70± 0.78 | 1.20± 1.40 | 1.30± 1.35 | 2.00± 1.26 |

Table 8: Zero-shot testing performance on training robot types (Exp. III & IV)

| Alg. | Testing Robot Types | | | | | | | |
|---|---|---|---|---|---|---|---|---|
| | A | B | C | D | F | G | H | I |
| HCP-E | 92.00± 2.28 | 89.60± 3.01 | 98.60± 1.20 | 99.00± 0.63 | 96.70± 1.42 | 97.90± 1.64 | 99.30± 0.64 | 99.20± 0.60 |
| DDPG+HER | 1.30± 0.90 | 1.30± 0.89 | 1.60± 0.92 | 0.70± 0.46 | 0.20± 0.40 | 2.30± 1.18 | 0.90± 0.83 | 1.40± 0.92 |

Table 9: Zero-shot testing performance on training robot types (Exp. V & VI)

| Alg. | Testing Robot Types | | | | | | | |
|---|---|---|---|---|---|---|---|---|
| | A | B | C | D | E | F | G | I |
| HCP-E | 91.10± 2.77 | 95.90± 1.92 | 98.50± 1.50 | 84.89± 3.29 | 94.70± 1.85 | 92.00± 2.97 | 97.20± 1.32 | 94.20± 2.79 |
| DDPG+HER | 0.30± 0.46 | 1.90± 1.30 | 3.00± 1.26 | 0.00± 0.00 | 0.00± 0.00 | 0.00± 0.00 | 0.60± 0.66 | 0.00± 0.00 |

Table 10: Zero-shot testing performance on training robot types (Exp. VII & VIII)

| Alg. | Testing Robot Types | | | | | | | |
|---|---|---|---|---|---|---|---|---|
| | A | B | C | D | E | F | G | I |
| HCP-E | 88.60± 2.45 | 95.30± 2.00 | 98.90± 0.83 | 83.30± 3.49 | 81.30± 3.20 | 92.00± 3.13 | 89.40± 3.20 | 88.00± 4.54 |
| DDPG+HER | 3.30± 2.32 | 1.70± 1.00 | 0.00± 0.00 | 0.00± 0.00 | 0.00± 0.00 | 0.00± 0.00 | 0.00± 0.00 | 0.10± 0.30 |

Table 11: Zero-shot testing performance on training robot types (Exp. IX & X)

| Alg. | Testing Robot Types | | | | | | | |
|---|---|---|---|---|---|---|---|---|
| | A | B | C | D | E | F | G | H |
| HCP-E | 92.90± 3.59 | 95.90± 1.70 | 97.30± 1.10 | 90.90± 3.58 | 95.59± 1.43 | 94.60± 1.28 | 98.80± 0.60 | 97.10± 1.51 |
| DDPG+HER | 1.60± 1.20 | 2.30± 1.27 | 0.40± 0.66 | 0.00± 0.00 | 1.80± 1.54 | 0.00± 0.00 | 0.40± 0.49 | 0.00± 0.00 |

Table 12: Zero-shot testing performance on training robot types (Exp. XI & XII)

| Alg. | Testing Robot Types | | | | | | | |
|---|---|---|---|---|---|---|---|---|
| | A | B | C | D | E | F | G | I |
| HCP-E | 71.00± 5.22 | 85.80± 3.19 | 89.00± 2.53 | 6.70± 2.53 | 79.30± 3.90 | 45.70± 4.86 | 88.50± 2.42 | 68.50± 5.18 |
| DDPG+HER | 1.70± 1.88 | 3.90± 2.20 | 0.90± 1.04 | 0.10± 0.30 | 2.50± 0.92 | 0.10± 0.30 | 1.00± 1.00 | 0.70± 0.78 |

Table 13: Zero-shot testing performance on training robot types (Exp. XIII & XIV)

| Alg. | Testing Robot Types | | | | | | | |
|---|---|---|---|---|---|---|---|---|
| | A | B | C | D | F | G | H | I |
| HCP-E | 64.70± 6.30 | 86.10± 3.36 | 89.60± 2.95 | 54.10± 3.53 | 58.60± 3.83 | 83.20± 2.96 | 66.30± 3.57 | 62.50± 4.03 |
| DDPG+HER | 0.30± 0.46 | 0.10± 0.30 | 1.90± 0.94 | 0.00± 0.00 | 0.20± 0.40 | 1.90± 1.30 | 0.10± 0.30 | 2.80 ± 1.60 |

Table 14: Zero-shot testing performance on training robot types (Exp. XV & XVI)

| Alg. | Testing Robot Types | | | | | | | |
|---|---|---|---|---|---|---|---|---|
| | A | B | C | D | E | F | G | H |
| HCP-E | 73.70± 4.79 | 86.20± 4.04 | 80.90± 3.88 | 16.00± 3.26 | 69.70± 4.54 | 76.80± 3.25 | 85.80± 4.38 | 59.90± 5.22 |
| DDPG+HER | 1.90± 1.37 | 7.60± 2.65 | 3.10± 1.04 | 0.00± 0.00 | 3.70± 1.62 | 0.60± 0.66 | 1.30± 1.00 | 0.40 ± 0.80 |