[Reviews · NeurIPS 2018]

Reviewer 1



Disclaimer: my background is in control theory and only recently I have invested most of time in reading and doing research in the area of machine learning and reinforcement learning with specific focus on robotics and control. I went through the submitted paper carefully, including the supplementary material. Therefore I am quite confident with my assessment, especially since the problem that the addressed problem is well inside my core expertise (adaptive control). As I previously said, I am very confident with the problem, less confident with the theoretical framework (reinforcement learning) used to solve it. The math presented in the paper is relatively shallow and carefully checked. I am quite confident with the used reinforcement learning tools (PPO, DDPG, DDPG+HER). = MAJOR COMMENTS = 1. The problem addressed in the paper is interesting. The very interesting part of the paper is the implicit encoding which proposes a solution to condition the control policy on quantities that are not directly measurable (e.g. dynamic properties). In the area of robotics, a similar problem has been solved with adaptive control (Slotine 1988, “Adaptive manipulator control: A case study”) where dynamic parameters (link masses, inertias and centers of mass, joint damping and friction) are estimated on-line using their linearity. The mathematical details of the proposed implicit encoding are confined in a relatively short subsection (Section 4.2 "Implicit encoding"). This section should be expanded and presented in depth. Details on how the cost function $L$ is constructed should be added so that the reader can appreciate what is novel with respect to (Slotine 1988). - The paper would be very difficult to replicate due to the lack of details. Here is a list of things that should be better described/detailed in order to improve the paper repeatability. Page 3. Section 4 (first paragraph). What is the structure of the policy network? What is the state $s_t$ (e.g. joint positions and velocities)? Page 4. Section 4.2. What is the cost function $L$? What is the vector of parameters $\theta$? This is fundamental to understand how $v_h$ is estimated from the available measurements ($\theta$?). = MINOR COMMENTS = Appendix B, Table 2. How is the length of a link measured (e.g. how are these parameters related to the Denavit-Hartenberg parameters)? What is the meaning of the armature of a joint? Page 2, Section 2 "Related work". A section on adaptive control for robotics should be added. A source of references can be found in (Slotine 1988) and in the book "Applied nonlinear control" by Jean-Jacques Slotine 1991. Page 3, Section 4.1 "Explicit encoding". The adopted explicit encoding is quite subjective and information-dense. It is subjective in the sense that the position of the local coordinate systems is totally arbitrary and the proposed results might depend on the specific choice of these coordinate systems. It is information-dense because it includes all the coordinate systems across the kinematic chain. It would be good to test how the proposed approach behaves when reducing this information, possibly reducing it to the end-effector pose only.

Reviewer 2



This paper presents a method for transfer learning between robots with various kinematics and dynamics. The main idea is to concatenate the state vector with a "hardware representation vector". The hardware representation vector is based on the physical structure and properties of each robot. The authors study explicit hardware representations vectors (based on the kinematics) and implicit (learned) ones (by back-propagating through the policy network). Strong points: - Works with existing RL methods (just concatenate the state vector with the hardware representation vector) - Seems easy enough to implement - Extensive experiments in simulation Weak points: - No experiments on real hardware experiments - Learning the implicit hardware embedding seems a bit naive. As it stands, this needs to be done for each new robot. How does this scale to few-shot or zero-shot learning? Questions: - Can you comment on the relationship between your work and this recent paper: https://arxiv.org/abs/1806.01242 . The architecture is very different, but both works claim to embed/learn physics related properties and allow for transfer learning. - Do you have any insights in the implicit embedding? What do the implicit hardware representation vectors look like (e.g. do they similar for similar robots)? The hardware representation vectors are a function of the robot's physical structure (h_v = f(robot)). Can this function f somehow be learned?

Reviewer 3



This paper presents an interesting development and demonstration of applications of deep learning of policies to transfer between heterogeneous hardware. The paper derives an interesting deep learning of policies which are novel work and interesting to the reader. The extra material presented is useful and serves to underpin the findings in the paper. It should be made available with the paper, although suits as extra material so as not to disrupt the flow. The video material is especially appreciated. The paper is technically sound and has produced good data, I do wonder how re-creatable said data is, it may be beneficial for the authors to present this more fully and make available for others in the field. However, one point I find interesting is that peg positions could be selected out of the range of the robots, this seems counter-intuitive for training purposes, as the setup makes a solution impossible. I do question whether this could have then have weighted the results in some manner, and it may be beneficial adding a discussion on this point. I've got concerns over how well the results would transfer to a real-world system, there are a collection of high frequency dynamics that could cause issues. There also look to be some collisions with the workspace which would present a potential problem. This should be investigated further. It should also be evaluated whether there is a wide enough range of training data, and whether this 70% represents all the reachable environment.